# Gender-Dependent Specificities in Cutaneous Melanoma Predisposition, Risk Factors, Somatic Mutations, Prognostic and Predictive Factors: A Systematic Review

**DOI:** 10.3390/ijerph18157945

**Published:** 2021-07-27

**Authors:** Oriana D’Ecclesiis, Saverio Caini, Chiara Martinoli, Sara Raimondi, Camilla Gaiaschi, Giulio Tosti, Paola Queirolo, Camilla Veneri, Calogero Saieva, Sara Gandini, Susanna Chiocca

**Affiliations:** 1Department of Experimental Oncology, IEO—European Institute of Oncology IRCCS, 20139 Milan, Italy; Oriana.Decclesiis@ieo.it (O.D.); chiara.martinoli@ieo.it (C.M.); sara.raimondi@ieo.it (S.R.); 2Cancer Risk Factors and Lifestyle Epidemiology Unit, Institute for Cancer Research, Prevention and Clinical Network (ISPRO), Via Cosimo il Vecchio 2, 50139 Florence, Italy; s.caini@ispro.toscana.it (S.C.); c.saieva@ispro.toscana.it (C.S.); 3GENDERS Center, Department of Social and Political Sciences, Università degli Studi di Milano, 20122 Milan, Italy; camilla.gaiaschi@unimi.it (C.G.); camilla.veneri@unimi.it (C.V.); 4Faculty of Social and Political Sciences, Institute of Social Sciences, University of Lausanne, 1015 Lausanne, Switzerland; 5Division of Melanoma Surgery, Sarcoma and Rare Tumors, IEO—European Institute of Oncology IRCCS, 20139 Milan, Italy; giulio.tosti@ieo.it (G.T.); paola.queirolo@ieo.it (P.Q.)

**Keywords:** melanoma, gender, sex, risk factors, prognostic factors, predictive factors, meta-analysis, systematic review, precision medicine

## Abstract

Background and aim: Over the last decades, the incidence of melanoma has been steadily growing, with 4.2% of the population worldwide affected by cutaneous melanoma (CM) in 2020 and with a higher incidence and mortality in men than in women. We investigated both the risk factors for CM development and the prognostic and predictive factors for survival, stratifying for both sex and gender. Methods: We conducted a systematic review of studies indexed in PUB-MED, EMBASE, and Scopus until 4 February 2021. We included reviews, meta-analyses, and pooled analyses investigating differences between women and men in CM risk factors and in prognostic and predictive factors for CM survival. Data synthesis: Twenty-four studies were included, and relevant data extracted. Of these, 13 studies concerned potential risk factors, six concerned predictive factors, and five addressed prognostic factors of melanoma. Discussion: The systematic review revealed no significant differences in genetic predisposition to CM between males and females, while there appear to be several gender disparities regarding CM risk factors, partly attributable to different lifestyles and behavioral habits between men and women. There is currently no clear evidence of whether the mutational landscapes of CM differ by sex/gender. Prognosis is justified by a complex combination of phenotypes and immune functions, while reported differences between genders in predicting the effectiveness of new treatments are inconsistent. Overall, the results emerging from the literature reveal the importance of considering the sex/gender variable in all studies and pave the way for including it towards precision medicine. Conclusions: Men and women differ genetically, biologically, and by social construct. Our systematic review shows that, although fundamental, the variable sex/gender is not among the ones collected and analyzed.

## 1. Introduction

Melanoma is a type of cancer that originates from melanocytes, specialized cells that produce the pigment melanin and are found in the skin, eye, and mucosal epithelia. Cutaneous melanoma (CM) is the most common type of melanoma, with different risk, prognostic, and predictive factors than melanoma originating from other tissues (e.g., ocular and mucosal melanomas).

In recent decades, CM incidence has been growing in the general population, with higher rates observed in North America than in Europe. CM affected an estimated 4.2% of the world population in 2020, with a higher incidence in men than in women (crude rate (CR) = 4.4 and CR = 3.9, respectively) [1]. In particular, the incidence was higher in women than in men up to 24 years of age, after which higher rates prevailed in men. The estimated mortality rate from CM for 2020 was also higher in men than in women (CR = 0.82 and CR = 0.64, respectively) and in Europe compared to North America (CR = 3.5 and CR = 2.3, respectively).

Differences in health, disease incidence, and outcomes between women and men depend on both their sex and their gender differences. The former refers to biological characteristics that derive from the binary classification of reproductive organs and from genetic differences, the latter to a set of roles, expectations, relationships, behaviors and other traits, and meanings that societies ascribe to women and men [2,3]. However, research has not always considered sex/gender disparities. This systematic review aims at summarizing the available evidence regarding similarities and possible differences between males and females in terms of CM genetic predisposition, risk factors, frequency of somatic mutations, and prognostic and predictive factors by examining all relevant reviews, meta-analyses, and pooled analyses that reported stratified results by sex/gender. Specifically, the scientific questions that this review aims to answer are the following: are there any differences between men and women regarding CM predisposition and risk factors? Does the CM mutational landscape differ in a systematic way between men and women? Can sex/gender be considered a prognostic factor for melanoma? Are there distinct prognostic factors for melanoma between sexes/genders? Finally, can sex/gender predict the effectiveness of new melanoma treatments?

## 2. Materials and Methods

### 2.1. Search Strategy

The literature search was conducted by using PubMed, EMBASE, and Scopus, until 4 February 2021, using the following search strategy TITLE-ABS-KEY (((sex AND factors) OR (gender*) OR (sex) OR (male) OR (female) OR (woman) OR (man) OR (women) OR (men)) AND ((genetic* AND gene* AND polymorphism* AND mutation*) OR (risk AND factors) OR (risk)) AND (melanoma) AND ((meta AND analysis) OR (review)) AND ((prognostic AND factor) OR (prognostic))). To include as many eligible studies as possible, no restrictions of time and language were imposed. Further studies have been included from the bibliography of relevant articles.

### 2.2. Inclusion and Exclusion Criteria

Studies were included if they met the following criteria: (a) studies focusing on CM; (b) review, meta-analysis, or pooled analysis that took sex/gender into account; and (c) articles dealing with CM genetic predisposition, risk factors, frequency of somatic mutations, or prognostic or predictive factors for CM. During the initial screening, publications were excluded if (d) they did not report summary estimates stratified by sex/gender. Criterion (b) was taken into consideration as it was essential to assess the level of awareness regarding the sex/gender difference issue in the searched literature and to limit the publication bias of the single study. Papers that focused only on one sex/gender were not included as the objective of our review was to compare estimates between the sexes/genders. No attempt was made to contact the authors of the original publications. The limitations disclosed by the authors of each study were discussed, including the presence or absence of publication bias.

### 2.3. Study Selection

Study selection of this review was conducted in two steps by one author (OD). First, titles and abstracts of all studies were screened for their relevance. Reasons for excluding studies at this stage were: unrelated topics; study type other than meta-analysis, review, or pooled analysis; no reporting of results stratified by gender. Second, the full text of the remaining studies was further screened, and eligibility verified.

### 2.4. Data Extraction and Quality Assessment

A researcher (OD) extracted data and the results were discussed with senior investigators until a consensus was reached. The outcomes used in the various studies were: CM incidence, overall survival (OS), progression-free survival (PFS), disease specific survival (DSS), time to lymph node metastasis (TLNM), time to distant metastasis (TDM), and relapse-free survival (RFS). The following data were extrapolated from each article: first author name, year of publication, journal, type of article, study design, number of studies included in the calculation of the summary estimate, geographic area, outcome, exposure, gender, summary estimate (any of odds ratio (OR), relative risk (RR), hazard ratio (HR), relative excess risk (RER) or population attributable risks (PAR), effect size (ES), or standardized incidence ratio (SIR)) with corresponding 95% confidence interval (CI) and the heterogeneity index (I^2^). Because melanoma incidence is low, we ignore the distinction between RRs and ORs.

The evaluation of the quality of the systematic review was made through the 27 items proposed by the PRISMA 2009 Checklist (Appendix A).

### 2.5. Analysis

The data were qualitatively summarized in narrative form. No meta-analysis was conducted due to the heterogeneity of the studies. The summary of findings focused on CM risk, prognostic and predictive factors relative to sex/gender.

## 3. Results and Discussion

### 3.1. Study Characteristics

As shown in Figure 1, the literature search produced 2031 publications of which 73 duplicates were discarded, plus an additional 80 articles identified in the reference lists. From the initially identified 2038 articles, 1866 were removed upon title and abstract screening, and 148 articles were excluded for not meeting the inclusion and exclusion criteria. A total of 24 studies were finally included in the present systematic review, and relevant data were extracted. Of these, 13 studies concerned potential risk factors, six concerned predictive factors, and five addressed prognostic factors of melanoma.

### 3.2. Risk Factors for Melanoma Stratified by Sex/Gender

Genetic factors: Germline mutations for only a few genes are known to be associated with an increase in sporadic CM risk. The *MC1R* (melanocortin 1 receptor) is the most important gene found to play a role in predisposition to sporadic CM, since several of its allelic variants show non-synonymous mutations, some of which convey a not-negligible increase in CM risk and are highly prevalent, although with substantial geographical variability [4]. To date, no systematic review has examined whether males and females differ in the prevalence of *MC1R* variants related to their CM risk, and only a few studies have explicitly addressed this topic. In a Spanish hospital-based case-control study, authors reported that *MC1R* variants associated with the red hair color phenotype had a more substantial effect on skin pigmentation among females than among males [5]. In a more recent study conducted in Austria, *MC1R* variants were found to convey an increased CM risk among females but not among males, upon adjusting for the history of sunburns and the presence of actinic skin damage [6]. In general, however, the existence of gender specificities in the association between *MC1R* variants and CM risk has been overlooked in most studies examining this topic, and most findings have not been corroborated in subsequent studies.

Two other well-known melanoma susceptibility genes widely investigated in melanoma-prone families are *CDKN2A* and *CDK4*. For these genes, as well, no systematic review and meta-analysis were identified. Still, unlike *MC1R*, several studies examined whether gender disparities exist in the association between germline mutations in these two genes and CM risk, and findings were fairly concordant in showing the absence of any interaction, e.g., concerning gene penetrance and clinicopathological characteristics of melanomas arising among carriers [7,8,9,10,11].

Kocarnik et al. examined the overall and gender-specific association of polymorphisms at several genes (previously reported to be associated with cancer risk at other body sites in genome wide association studies) with CM and identified a single nucleotide polymorphism between the *TPCN2* and *MYEOV* genes that was marginally associated with CM risk only among male carriers. At the same time, no significant associations have emerged between sex and polymorphisms in other genes, including *TERT/CLPTM1L* [12]. In conclusion, there is only limited evidence that genetic predisposition to CM (<7% of all CM cases are attributable to familial risk [13]) differs by sex. However, the studies examining this topic are fairly limited in number to date.

Tanning devices: Using tanning devices increases the risk of non-melanoma [14] and melanoma skin cancers [15]. A recent meta-analysis conducted in Canada ([16], Table 1) showed that, in 2015, 7.0% of melanomas were attributable to constant use of indoor tanning devices, with a higher percentage in females (10.1%) compared to males (4.6%). Interestingly, in females, the estimate of risk attributable to the population for melanoma associated with indoor tanning decreased with age, while in males it remained constant with a peak in the age category 25–44 years. The authors indicated among the limits the use of a single Canadian study to estimate the exposure. It should be noted that the difference in terms of the incidence of melanoma between males and females also depends on age. In all countries, females have higher rates of melanoma than males during early childhood, while males have higher rates of melanoma than females in later life [17]. The topic of “indoor tanning” is much debated and mainly concerns young adults. It is among these subjects that most of the burden falls [18]. Both the activation of targeted campaigns and banning the use of tanning devices to adolescents would be helpful. Rodriguez-Acevedo et al. [19] in their study emphasized the possible effects of laws prohibiting indoor tanning to adolescents, in terms of prevalence in the use of tanning devices. Indeed, as expected, they showed that the use of indoor tanning was higher in countries that did not adhere to this type of legislation than in countries that, on the contrary, adopted this type of legislation and that this difference was greater among females than males. Furthermore, there was a significant difference in female adolescents in countries with a ban compared to those in countries without any prohibition (4.2% vs. 8.7%).

Two systematic reviews investigated the prevalence of skin cancer and skin cancer risk behaviors among sexual and gender minority populations [20,21]. They found that sexual minority men (SMMs) have a higher risk of melanoma over their lifetime than heterosexual men (OR = 1.7, 95% CI 1.1–2.7) [22]. This can be explained by the fact that they may be more at risk of indoor tanning than heterosexual men [23]. Both aesthetic concerns and community pressures are probably related to indoor tanning use [21]. Similarly, among young women, indoor tanning is strongly associated with psychological and social factors (e.g., body image disorders, depression), which may be related to the motivations for exposure to UV radiation [21,24,25,26]. In addition to young girls, women also tend to use indoor tanning: in particular, women belonging to sexual minorities use indoor tanning less frequently than heterosexual women [27] and do not have a high risk of a lifelong history of non-melanoma skin cancer [22]. However, no differences emerged in terms of melanoma risk between heterosexual women and those belonging to sexual minorities [22]. A limitation of both reviews is that in the original studies, most of the data were self-reported.

Patients from gender minorities, including trans women who receive feminizing hormone therapy with estrogen derivatives, may have an increased risk of melanoma [20,21]. However, this hypothesis is still controversial since data showing an association between estrogen exposure and melanoma are still unclear [21,28].

Obesity and height: Obesity is a risk factor for many types of cancer [29,30]. A meta-analysis of cohort studies ([31], Table 1), which examined the association between obesity and the risk of melanoma, detected a significantly increased risk of melanoma in overweight and obese men (OR = 1.29, 95% CI 1.15–1.45, and OR = 1.30, 95% CI 1.17–1.45, respectively) but not among women (OR for overweight = 0.99, 95% CI (0.92–1.07), OR for obesity = 0.87, 95% CI 0.70–1.08) when body-mass index (BMI) was used as an indicator. Interestingly, after adjusting for sunlight exposure, these relationships changed: no significant association was detected for males (OR = 1.74, 95% CI 0.86–3.58), while for females, overweight/obesity was significantly associated with reduced melanoma risk (OR = 0.80, 95% CI 0.64–0.99). Moreover, when obesity was measured in terms of body surface area (BSA), a significant positive association between obesity and melanoma risk was observed in men (OR = 1.84, 95% CI 1.39–2.45) but not in women (OR = 1.37, 95% CI 0.94–2.00). When adjusting for sunlight exposure, in males there was no change, while in women, the association became significant (OR = 1.51, 95% CI 1.04–2.19). Of note, if we think in terms of the surface exposed to the sun, the risk of melanoma increases in both sexes, especially in men, contrary to what happens in women, thinking in terms of obesity. Often obese women, probably because of social convention, do not expose themselves to the sun or expose themselves less than obese men. A residual confounding may also be present, and it may be significant in cohort studies as, for example, some studies determined sunlight exposure only at baseline and/or at the end of the study. Thus, all changes in sunlight exposure during the follow-up could not be accurately taken into account. Since sun exposure at a young age seems important, it may be a problem not to consider that both obesity and sun exposure can change throughout life [31]. This indicates the need for these types of studies to consider possible confounding factors. Among the discussed limitations of these studies, the main concern is that not all estimates were adjusted by sun exposure. Furthermore, as many studies have shown that the skin phototype is an important risk factor for melanoma, the adjustment for this variable would have made the results more reliable.

In line with these findings, another meta-analysis ([32], Table 1) showed a significant positive association between 5 kg/m^2^ increase in BMI and melanoma risk (OR = 1.17, 95% CI 1.05–1.30) in males, while no significant association was detected for women (OR = 0.96, 95% CI 0.92–1.01). This study, however, did not take into account many factors that could act as confounders, including the self-reported weight, which was lower than actual body weight [33]. In turn, the underestimation of weight increases in older individuals [32,34].

A systematic review and meta-analysis of prospective cohort studies aimed at estimating the relationship between height and risk of melanoma ([35], Table 1). Stratified gender analysis showed that ORs of melanoma risk associated with a 10 cm increase in height in men and women were 1.52 (95% CI: 1.01–2.29) and 1.27 (95% CI: 1.18–1.36), respectively. Most of the studies were conducted in North America and Europe; therefore, the results cannot be generalized to other countries.

Thus, to summarize, exposure to ultraviolet radiation from the sun is the leading proven cause of skin melanoma [36,37,38]. The association with obesity and height could be related to the increased surface area of the skin that can be exposed to sunlight. One possible explanation for the fact that women have lower estimates than men for the risk factors mentioned above is that women may tend to expose less to the sun and cover themselves more if obese or if, in general, they do not feel comfortable with their body.

Alcohol, smoking, and coffee consumption: For various cancers, alcohol consumption is an important and preventable risk factor [39]. Several studies have suggested that alcohol intake may be associated with sunburn severity, a significant risk factor for skin melanoma [40]. Indeed, the consumption of alcoholic beverages often occurs during outdoor activities, especially in Western societies [41]. A meta-analysis of cohort and case-control studies investigated the association between alcohol consumption and melanoma risk in subgroup analyses according to gender ([42], Table 1). No significant association emerged for males (OR = 1.47, 95% CI: 0.94–2.29), while females who regularly drink alcohol, compared to women who do not drink alcohol or occasionally drink it, showed a 26% higher risk of melanoma (OR = 1.26, 95% CI: 1.19–1.35). Although the risks by gender were not statistically different (*p* = 0.41), an explanation for the results could be that females who drink alcohol are also more prone to seek tanning for aesthetic reasons than males [43], resulting in a possible increased risk of melanoma. Importantly, the results should be interpreted with caution as the observed association derives just from 6 out of 16 works considered in the original review [42]. Therefore, the results should be interpreted with caution as residual confounding by sun exposure cannot be ruled out.

It is now known that smoking is a severe risk factor, along with alcohol, for many types of cancer [44]. A meta-analysis of cohort studies on healthcare professionals conducted by Fengju Song et al. ([45], Table 1) studied the relationship between smoking and melanoma risk. Men smokers were found to have a 27% lower risk of melanoma (OR = 0.73, 95% CI 0.67–0.80) compared to non-smokers, while no statistically significant association between smoking and melanoma risk was detected in women (OR = 0.95, 95% CI 0.85–1.05). A limitation highlighted by the authors is the possible presence of detection bias due to the smoking status. Although the inverse association between smoking and melanoma risk is not fully understood, some authors suggested [46] that tobacco may exert an immunosuppressive effect that would protect against deleterious immune reactions caused, for example, by the sun’s rays. Another possible mechanism, as discussed in [45], is that smoking can decrease Notch pathway gene expression, described to enhance melanoma cells’ growth [47]. Furthermore, smoking increases elastosis, which can reduce the development of melanoma [48]. Several studies have shown that the consumption of coffee can elicit a protective effect in the development of melanoma [49]. In addition to the evidence from epidemiological studies that coffee consumption reduces the risk of colon and liver cancer, various bioactive compounds found in coffee can target different paths of UV-mediated melanoma carcinogenesis [49,50,51,52,53]. A systematic review and meta-analysis of observational studies evaluated associations between coffee consumption and risk of melanoma [49]. No significant association emerged for men (OR = 1.22, 95% CI 0.83–1.77), while women who drink coffee regularly, compared to women who do not consume or consume it occasionally, showed a 38% lower risk of melanoma (OR = 0.62, 95% CI 0.47–0.82). However, the difference between men and women was not statistically significant (*p* = 0.33). The content of polyphenols in coffee contributes to its antioxidant capacity [54]. From many studies, it emerged that females develop more effective immune responses than males [55], partly explaining the results. However, the interpretation is best to be cautious since several potential confounding factors, not appropriately considered in the study, could affect the observed associations (e.g., coffee type, coffee processing, geographical area, phenotypical factors, and sun exposure).

Parkinson’s disease: Epidemiological evidence supports an association between Parkinson’s disease (PD) and melanoma [56,57,58]. Many studies revealed that PD patients are subject to a lower incidence of many types of cancer but not melanoma [58,59]. The results of a meta-analysis ([58], Table 1), which took gender into account, showed a positive association between PD and melanoma in both men (2.04, 95% CI 1.55–2.69) and women OR = 1.52, 95% CI 0.85–2.75), although in the latter it was not statistically significant. In accordance, another meta-analysis ([60], Table 1) also described a statistically significant association between PD and melanoma in both men (OR = 1.64, 95% CI 1.27–2.13) and women (OR = 1.38, 95% CI 1.04–1.82), although the associations were weaker in the latter. The limitations highlighted for these two meta-analyses were similar. The number of cases was often small, and most of the included studies did not focus on the risk of melanoma in patients with PD. Co-occurrence of PD and melanoma could be at least in part explained by polymorphisms in melanocortin-1-receptor (*MC1R*) gene [61]. Loss of function variants of *MC1R* increases the risk of melanoma as well as PD [61]. The slight disparity between men and women in the strength of the estimated association could be caused by the fact that, compared to men, women have a lower risk of both melanoma and PD [62,63].

Other factors: Robert Dellavalle et al. investigated the relationship between the use of statins and the risk of melanoma in men and women ([64], Table 1). There was no significant association between statin use and melanoma risk in either group (ES for men = 0.88, 95% CI 0.54–1.44; ES for women = 0.94, 95% CI 0.59–1.51, respectively). We must be careful with the interpretation since the 13 studies included in the review mostly concerned men. Furthermore, in most of the studies, cancer was not the primary outcome, and not all data on participants were available regarding the diagnosis of melanoma.

From a review [65] assessing whether there was a chance relationship between melanoma and ionizing radiation, we extracted information from two studies. A retrospective cohort study ([66], Table 1) investigated the cancer incidence in a specific category of workers, the Nordic airline cabin crew. A higher incidence of melanoma was observed in both women and men (SIR for women = 1.85, 95% CI 1.41–2.38; SIR for men = 3.00, 95% CI 1.78–4.74). The study by P. Reynolds et al. ([67], Table 1) on the cancer incidence in California flight attendants is consistent with these results. Here, the estimates stratified by sex were SIR for men = 3.93, 95% CI 0.74–11.6; SIR for women = 2.50, 95% CI (1.28–4.38). Several limitations emerged from this study; indeed, the fact that flight attendants are very frequent travelers makes it difficult to compare their risk estimate with that of the referent population, in this case California, since they could not be considered individual residents in a permanent place. Although many studies indicate that cabin and flight attendants are more exposed to cosmic radiation [68], which can increase the incidence of various cancers, other studies attribute a higher incidence of melanoma in these categories of workers because of their opportunity for traveling [69]. Indeed, one consideration could be that their touring to exotic sunny places can lead to increased sun exposure, a widely known risk factor for melanoma, as discussed above.

### 3.3. Somatic Mutations in known CM Driver Genes

The skin is constantly exposed to UV radiation, and CM has a very high somatic mutation rate [70,71]. Although the genes most frequently mutated in CM are known [72], there is still uncertainty as to whether the mutational landscape of melanoma differs between genders, also because the gender specificity (if any) may be partly masked and confounded by parallel differences between sexes in the patterns of exposure to natural and artificial UV. Overall, the recent literature on this topic is characterized by conflicting data. Lotz et al. fitted mathematical models of how mutations accumulate in cancer genomes and found that men are more susceptible than women to cell-division-linked (but not age-related) mutations because of either an inherently faster proliferation rate of male melanocytes or a decreased repair capacity of UVR-induced mutations [73]. An analysis conducted among 266 patients with metastatic melanomas (38.3% were women) from The Cancer Genome Atlas (TCGA) found that only missense, nonsense, stop loss, and frameshift mutations were more frequent among men [74], while the number of “UV signature” mutations (i.e., C✍T or CC✍TT changes) did not differ between men and women. This finding, according to the authors, ruled out UV exposure differences as the sole explanation of their findings and pointed towards an actual disparity between sexes, which might be explained by a less effective antitumor immune surveillance among men. Somewhat in contrast with these findings, two meta-analyses published in 2011 and 2015 did not find any significant differences between sexes in the proportion of BRAF or NRAS mutations (except for a 15% reduced proportion of BRAF mutations among men in white populations only, reported by Kim et al.). Likewise, a more recent (2020) systematic review found no differences between sexes in the frequency of mutations in BRAF, NRAS and KIT genes (based on 27, 15, and 8 articles, respectively) [75]. In conclusion, the question remains open and new studies are warranted, also considering the prognostic value of the presence of mutations in these three genes.

### 3.4. Sex/Gender as a Potential Predictive Factor for Melanoma

The immune system plays a crucial role in regulating the spread of cancers: indeed, an efficient immune response has the potential to eliminate cancer cells. However, many studies have shown that cancers can escape the immune response by engaging immune checkpoints [76,77], as underlined by the emerging notion of anti-cancer agents capable of modulating them (immune checkpoint inhibitors, ICIs) [78,79]. We identified four studies focused on ICI therapy that evaluated whether sex could act as a predictor. Grassadonia et al. ([80], Table 2) explored, through a meta-analysis of randomized phase III clinical trials, the impact of sex on survival of advanced cancer patients treated with anti-CTLA-4, an ICI. The results showed that male melanoma patients obtained a significant benefit from anti-CTLA-4 therapy in terms of overall survival (OS) (HR = 0.67, 95% CI 0.50–0.90), although there was significant heterogeneity between studies (I^2^ = 77%). A similar, although slightly less pronounced, benefit from anti-CTLA-4 over standard therapy was observed for women (HR = 0.80, CI% (0.68–0.94)).

In agreement with these findings, the meta-analysis conducted by Conforti et al. ([81], Table 2) also reported improved survival in melanoma patients treated with ICIs compared to controls with a more pronounced effect in men than in women (HR for men = 0.66, 95% CI 0.55–0.79; HR for women = 0.79, 95% CI 0.70–0.90), with slight heterogeneity between studies in the male group (I^2^ = 60%). The difference between the two studies is that the meta-analysis by Grassadonia et al. only involved phase III studies, which guarantee a longer follow-up and a greater number of events, while that of Conforti et al. also included phase II studies.

In the meta-analysis ([82], Table 2) of randomized clinical trials of Christopher JD Wallis et al., which compared immunotherapy with standard therapy in the treatment of malignant neoplasms, no statistically significant difference between men and women was observed in melanoma survival (*p* = 0.36). However, a greater effect for men upon immunotherapy was suggested (HR for men = 0.68, 95% CI 0.48–0.97; HR for women = 0.83, 95% CI 0.68–1.00).

Yingchheng Wu et al. ([83], Table 2) conducted a meta-analysis investigating whether the efficacy of PD-1 and CTLA-4 inhibitors varied in males and females. With OS as endpoint, a significant benefit emerged in patients receiving ICIs, compared to controls, in both males and females (HR for men = 0.53, 95% CI 0.44–0.62; HR for women = 0.73, 95% CI 0.59–0.86), although a significant sex-related difference in efficacy was observed (*p* = 0.012) with a greater effect found in men. With progression-free survival (PFS) as endpoint, the estimates between males and females were not very different (HR for men = 0.52, 95% CI 0.40–0.64; HR for women = 0.56, 95% CI 0.39–0.72). In fact, the test on the difference between the two groups was not statistically significant (*p* = 0.70). While no publication bias emerged, the authors enumerated several study limitations, including limited access to PFS data, resulting in a lack of more detailed evaluation of the results. Furthermore, no safety profile was available regarding sex.

One hypothesis to explain why, among patients treated with ICIs, men show slightly greater benefits than women can be found in the sex-related differences of the immune system at baseline, as indicated by the higher number of CD8+ T lymphocytes and the lower CD4+/CD8+ ratio in males than in females [84]. Furthermore, the higher amount of T regulatory cells (Treg) in men, the subpopulation preferentially depleted by anti-CTLA-4 antibodies, could be a factor influencing the different responses, thus, producing a more substantial male benefit ([85]. Assuming that women generally have more robust immune systems than men, it is reasonable to expect diverse sex-associated anti-tumor effects of ICIs [86]. Nonetheless, enrolling a balanced number of men and women in clinical trials is of utmost importance to better understand sex differences and achieve the best healthcare for all [87].

A meta-analysis ([88], Table 2) of RCT studies evaluated interferon-α (IFN- α) as adjuvant therapy for high-risk melanoma treatment. The primary endpoint was event-free survival (EFS), which is the period of time after treatment during which the patient remains free from cancer-related events. Estimates, stratified by sex, compared IFN-α-treated patients versus controls and showed a benefit for IFN-α-treatment both in men (HR = 0.89, 95% CI 0.79–0.99) and women (HR = 0.88, 95% CI 0.79–0.99). No significant difference was observed between males and females (*p* = 0.90). The authors acknowledge that publication bias cannot be excluded.

PD-L1 is one of the checkpoints involved in modulating the immune response. Treatment with drugs that inhibit the PD-L1 checkpoint allows the elimination of these inhibitory pathways, thus, restoring the antitumor activity of T cells. From a pooled analysis ([89], Table 2) involving 1062 patients, conducted to identify the prognostic role of PD-L1 expression in melanoma, a non-significant association between PD-L1 expression and sex emerged (OR = 1.29, 95% CI 0.90–1.84; *p* = 0.16).

### 3.5. Sex/Gender as a Potential Prognostic Factor for Melanoma

Many studies demonstrated that women affected by melanoma have a better prognosis than men [90,91,92,93,94,95]. A study ([96], Table 3) on the Swedish population (N = 711) showed that female sex is associated with a better prognosis than male sex (OR = 0.80, 95% CI 0.80–0.90; *p* = 0.003). This result is in line with the findings of a pooled analysis conducted by Manola et al. ([94], Table 3) on patients with metastatic melanoma (OR = O.87, 95% CI 0.77–0.98; *p* = 0.02). A limitation that emerged from this study is that the inclusion of non-randomized phase II studies can produce heterogeneity, which can negatively affect the analysis. On the one hand, several studies showed that female sex is an independent prognostic factor [97]. As an example, a study [98] conducted in the Netherlands on 10.538 melanoma patients showed that men had a higher relative risk of death compared to women after adjusting the risk estimate for several variables (RER of dying 1.87, 95% CI 1.65–2.10). Other studies showed that better prognosis in women correlated with the tendency to present with a diagnosis of thin, non-ulcerated melanomas located on the extremities, while men generally have thick, ulcerated, and trunk melanomas [99]. Interestingly, a cross-sectional study showed that in males the appearance of melanomas on the trunk compared to the head and neck is typical of the younger age group (average 45 vs. 52.5 years), while the higher presence of melanoma in the lower limbs in females was already visible on average around 27.5 years of age [17]. The differences in terms of the anatomical site may be due both to behavioral factors, as women may generally take more precautions when exposed to the sun leading to better protection of the head and neck areas, and to biological factors, as it has been published that boys (6- to 7-year-old) had more moles on the head and neck than girls, with the upper and lower limbs being the most noticeable anatomical site [100]. The difference in the number of nevi ranging from 2.0 mm to 4.9 mm found in diverse body sites in children is consistent with the sex/gender disparity found in adults regarding the localization of melanoma in various sites of the body [101,102,103].

A pooled analysis, conducted by Jossee ([104], Table 3) on patients with stage I/II melanoma (N = 2672), showed that women have better OS (HR = 0.70, 95% CI 0.59–0.83), disease-specific survival (HR = 0.74, 95% CI 0.62–0.88), time to lymph node metastasis (HR 0.70, 95% CI 0.51–0.96), and time to distant metastasis (HR = 0.69, 95% CI 0.59–0.81) than men. Furthermore, the analysis found that, in general, women showed an advantage in terms of any aspect related to the progression of localized melanoma of approximately 30% compared to men. Another pooled analysis on 2734 stage III melanoma patients ([105], Table 3) showed that women had a 5-year disease-specific survival (DSS) rate higher than men (51.5%. vs. 43.3%) and a significantly better prognosis: DSS adjusted HR of 0.85 (95% CI 0.76–0.95), a relapse-free survival adjusted HR of 0.86 (95% CI 0.77–0.95), and an OS adjusted HR of 0.81 (95% CI 0.72–0.91). Finally, a pooled analysis of 1306 patients with stage VI melanoma [105] showed that women have an advantage in terms of DSS (HR = 0.81, 95% CI 0.72–0.92), PFS (HR = 0.79, 95% CI 0.70–0.88), and OS (HR = 0.82, 95% CI 0.72–0.93) compared to men. We note that in all three pooled analyses, the prognostic value of sex/gender was evident at each stage of the disease. However, in these pooled analyses the results cannot be generalized due to the inclusion criteria that led to a selected population since the data were retrieved from clinical trials.

Published literature, thus, shows that women with melanoma have a better prognosis than men, most likely due to both behavioral and biological reasons. Regarding the former, women generally lead a more balanced lifestyle and undergo more medical check-ups than men, who are less likely to seek medical attention or discover a non-advanced stage of cancer [106]. In fact, the probability of self-disclosure in women is 69%, while in men, it is 47% [107]. Regarding the latter, women have a more active immune system [108]. In addition, estrogen enhances the activity of the immune system while testosterone attenuates it [106].

A meta-analysis ([109], Table 3) conducted by Sai-Nan Han et al., investigating the prognostic role of the platelet-to-lymphocyte ratio (PLR) in melanoma, showed no significant association between PLR and sex (OR = 1.14, 95% CI 0.23–5.66; *p* = 0.87). There was no publication bias, although there may be selection biases and unacknowledged confounding factors typical of retrospective cohort studies.

## 4. Limitations

We acknowledge that some limitations in this review are based upon the difficulty of always separating gender from sex, including differentiating the cultural behaviors from biological impacts, which in many cases are interconnected [110]. The second is related to the fact that the sex/gender variable is most of the time measured by presuming a coincidence between the sex of an individual and her/his/their gender, not acknowledging trans and non-binary identities [111]. Furthermore, often sex/gender analyses are not reported in the article’s abstract, as they are considered secondary. Since our first screening was based on assessing the title and abstract for the presence of these analyses, this may have led to an incomplete collection of literature.

## 5. Conclusions

Historically, research and clinical studies were mostly conducted on men, with broadly generalizable diagnosis and ignoring the diversity among the sexes in human biology and physiology. Although it is now known that there are crucial differences between the sexes in many instances, not all research takes them into account during the design or analysis phase. Significant differences between men and women emerged concerning risk factors and prognostic factors for melanoma. Numerous risk factors affect melanoma in different ways between males and females, which in large part may be justified by different habits, behaviors, and lifestyles. Disregarding sex/gender disparity during the analysis phase could introduce bias. For example, in the studies reported in this review evaluating the association between obesity and melanoma risk, an interesting finding emerged. It is probably not so much obesity in terms of body fat that is a possible risk factor for melanoma, but it is a combination of sex/gender-related behavioral factors. Obese women compared to obese men tend to expose themselves less to the sun, the main risk factor for melanoma, due, for example, to social norms. Furthermore, it was found that the difference between women and men in terms of melanoma prognosis is given by a complex combination of factors, such as immune function and phenotypes. Regarding the sexual disparities in predicting the effectiveness of new treatments compared to standard ones, the results are not entirely consistent, and therefore, more in-depth studies are needed.

Finally and importantly, sex/gender differences in melanoma are crucial factors to consider in precision medicine.

## Figures and Tables

**Figure 1 ijerph-18-07945-f001:**
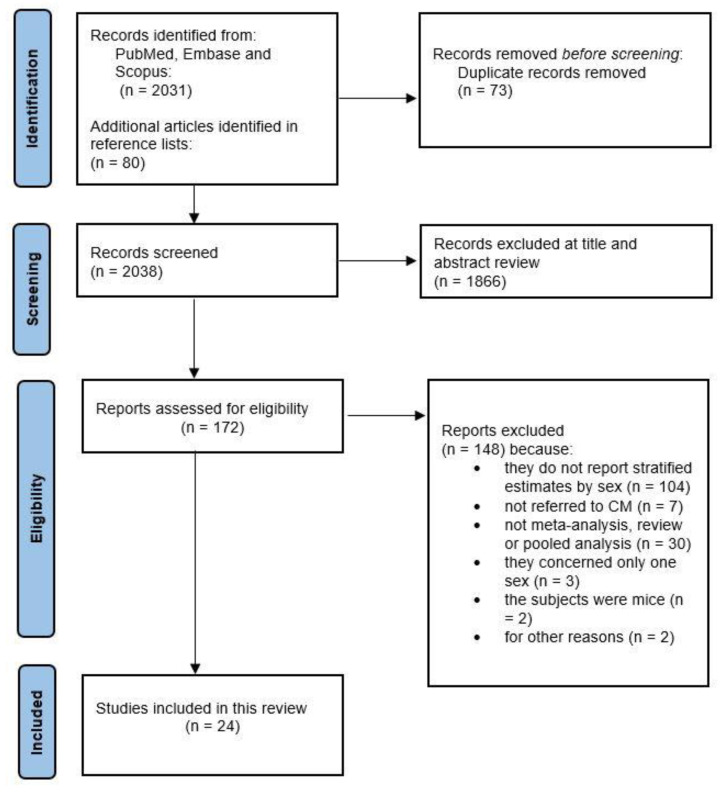
Flow diagram.

**Table 1 ijerph-18-07945-t001:** Studies evaluating the incidence of melanoma.

KERRYPNX	Article type	Study Designs	Exposure	Sex	N Studies	OR (95% CI)	$I^{2}$
O’Sullivan D.E. (2019)	MA	Co, CC, NCC	Indoor tanning	M W	8 8	PAR = 4.60 PAR = 10.10	
Rota M. (2014)	MA	Co, CC	Alcohol drinking	M W	3 3	1.47 (0.94–2.29) 1.26 (1.19–1.35)	45.7% 0%
Yik W. Y. (2015)	MA	Co, CC	Coffee consumption	M W	4 5	1.22 (0.83–1.77) 0.62 (0.47–0.82)	
Song F. (2012)	MA	NCC	Smoking	M W	4 3	0.73 (0.67–0.80) 0.95 (0.85–1.05)	57.3% 0%
Dellavalle R. (2009)	Rev	RCT	Statins	M W	3 2	0.88 (0.54–1.44) 0.94 (0.59–1.51)	
Fink C. A. (2005)	Rev	Co	Cabin attendants	M W		SIR = 2.9 (1.1–6.4) SIR = 1.7 (1.01–2.7)	
Fink C. A. (2005)	Rev	Co	Flight attendants	M W		SIR = 3.9 (0.74–11.6) SIR = 2.5 (1.28–4.38)	
Huang P. (2015)	MA	Co, CC, Cross	Parkinson Disease	M W	8 7	1.64 (1.27–2.13) 1.38 (1.04–1.82)	41.1% 45.3%
Liu R. (2011)	NA	Co, CC, NCC, Cross	Parkinson Disease	M W	8 5	2.04 ((1.55–2.69) 1.52 (0.85–2.75)	0% 54.5%
Sergentanis N.T (2012)	MA	Co	BMI: Overweight	M	5	1.29 (1.15–1.45)	35.6%
				W	5	0.99 (0.92–1.07)	23.4%
			Obesity	M	6	1.30 (1.17–1.45)	26%
				W	5	0.87 (0.70–1.08)	37.2%
			Both ^	M	1	1.74 (0.85–3.68)	
				W	2	0.80 (0.64–0.99)	0%
			BSA:				
			Obesity	M	5	1.84 (1.39–2.45)	31.0%
				W	7	1.37 (0.94–2.00)	46.8%
			Both ^	M		2.01	
				W		1.51 (1.04–2.19)	
Renehan A.G. (2008)	Rev	Co, NCC, CT	5 kg/m^2^ increase in BMI	M W	6 5	1.17 (1.05–1.30) 0.96 (0.92–1.01)	44% 0%
Yu D.J. (2018)	Rev	Co	10-cm increment in the height	M W	3 7	1.52 (1.01–2.29) 1.27 (1.18–1.36)	72.8% 59.5%
Singer S. (2020)	Rev	Cross	Sexual minority populations			1.70 (1.10–2.70)	

BMI, body mass index; BSA, body surface area; CI, confidence interval; M, men; NA, not available; OR, odds ratio; PD, Parkinson’s disease; PAR, population attributable risk estimate; RCT, randomized clinical trial; SIR, standardized incidence ratio estimate; W, women; I^2^, heterogeneity index; ^, adjusted for sunlight exposure; MA, meta-analysis; Rev, systematic reviews; Co, cohorts; NCC, nested case-control study; CC, case-control study; Cross, cross-sectional studies.

**Table 2 ijerph-18-07945-t002:** Characteristics of studies investigating predictive factors.

KERRYPNX	Article Type	Study Design	Exposure	Outcome	Sex	N Studies	HR (95% CI)	$I^{2}$
Grassadonia A (2018)	MA	RCT	ICIs (anti CTLA-4)	OS	M W	3 3	0.67 (0.49–0.89) 0.80 (0.68–0.93)	77% 0%
Wu Y. (2018)	MA	RCT	ICIs		M	5	0.53 (0.44–0.62)	
				OS PFS	W	5	0.73 (0.59–0.86)	
					M	5	0.52 (0.40–0.64)	
					W	5	0.56 (0.39–0.72)	
Conforti F. (2018)	MA	RCT	ICIs	OS	M W	7 7	0.66 (0.55–0.79) 0.79 (0.70–0.90)	60% 0%
Natalie J. Ives (2017)	MA	RCT	IFN-alpha	EFS	M W	15 15	0.89 (0.79–0.99) 0.88 (0.76–1.01)	
Christopher J. D. Wallis (2019)	MA	RCT	IO (immunotherapy)	OS	M W	4 4	0.68 (0.48–0.97) 0.83 (0.68–1.00)	
Yang J. (2020)	PA	RCT	Sex	PD-L1		7	1.29 (0.9–1.84)	30.3%

CI, confidence interval; CTLA, cytotoxic T-lymphocyte antigen; EFS, event free survival; HR, hazard ratio; ICIs, immune checkpoint inhibitors, IFN-alpha, interferons-alpha; M, men; MA, meta-analysis; OS, overall survival; PD-L1, programmed death-ligand 1; PFS, progression free survival; PA, pooled analysis; W, women; I^2^, heterogeneity index; RCT, randomized clinical trial.

**Table 3 ijerph-18-07945-t003:** Characteristics of studies investigating prognostic factors.

	Article Type	Study Design	Exposure	Outcome	Sample Size	OR (95% CI)	$I^{2}$
Sai Nan Han (2020)	MA	Co	sex	PLR	180	1.14 (0.23–5.66)	76.4%
Masback A. (2001)	Rev	Co	sex	OS	711	0.80 (0.80–0.90)	
Manola J. (2000)	PA	RCT	sex	OS	547	0.87 (0.77–0.98)	
				Stage I/II:	2672		
				OS		0.70 (0.59–0.83)	
				DSS		0.74 (0.62–0.88)	
				TLNM		0.70 (0.51–0.96)	
				TDM		0.69 (0.59–0.81)	
Jossee A. (2013)	PA	RCT	sex	Stage III:			
				DSS		0.85 (0.76–0.95)	
				RFS	2734	0.86 (0.77–0.95)	
				OS		0.81 (0.72–0.91)	
				Stage IV:	1306		
				DSS		0.81 (0.72–0.92)	
				PFS		0.79 (0.70–0.88)	
				OS		0.82 (0.72–0.93)	

CI, confidence interval; Co, cohorts; DSS, disease specific survival; OR, odds ratio; OS, overall survival; PFS, progression free survival; PLR, platelet to lymphocyte; RFS, relapse-free survival; TLNM, time to lymph node metastasis; TDM, time to distant metastasis; W, women; I^2^, heterogeneity index; PA, pooled analysis; MA, meta-analysis; Rev, systematic reviews; RCT, randomized clinical trial.

## Data Availability

Not applicable.

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
