# Peer review of "Gender-Dependent Specificities in Cutaneous Melanoma Predisposition, Risk Factors, Somatic Mutations, Prognostic and Predictive Factors: A Systematic Review"

_ijerph, 2021, doi:10.3390/ijerph18157945_

Round 1

Reviewer 1 Report

The proposed study is overall good and points out the needing of considering sex/gender among the variables determining the risk of developing cutaneous melanoma as well as immune therapy response. Also, the study shows how known risk factors for melanoma differently segregate among sexes. 

The reported data sound robust and statistical evaluation is well performed. However, some of the biological mechanisms proposed by the authors to explain the observed associations are poorly robust and in contrast with the pathophysiological and clinical features of melanoma. Aside from the statistical significance, associations should be always supported and justified by convincing biological mechanisms. Thus, the authors must improve the discussion for the work to be published. 

I reported below the points that should be addressed and improved by the authors, together with minor errors that should be amended to improve text quality: 

Line 70: it should be “by using”

Line 111: the authors provided only a Supplementary Table 1 and do not previously mention any other supplementary table. I guess it is a typo, but it should still be amended

Line 128: it should be “few genes”

Lines 130-131: this sentence is incorrect. If the presence of multiple alleles is documented, as it is, their existence is not a possibility, so the usage of “may” is incorrect in this case. Additionally, a gene carries mutations, not allelic variants. Therefore, the authors should rewrite the sentence amending these errors. If they want to emphasise the presence of non-synonymous mutations, an option could be “since several of its allelic variants show non-synonymous mutations, some of which confer a… etc”.

Line 134: it should be “few studies”

Lines 137-138: I recommend the authors to split the two sentences and related studies. From how it is written, it looks like the two studies have been performed by using the same patients’ cohort, while from the references it is clear that they are two separated studies

Line 144: ref. 7 only refers to CDKN2A variants. The authors might add it at the end of the paragraph (line 149), since also references n° 8-11 refer to melanoma-prone families.

Lines 150-153: The observed SNP is between TPCN2 and MYEOV genes, not within TPCN. It is not a trivial difference, since mutations within the gene affects the function of the protein, while mutations in the flanking regulatory regions generally affect levels of expression. Thus, the text should be amended accordingly. Additionally, Kocarnick et al specify that the association they observed is only marginally associated with CM risk in males, while from the text such association sounds statistically significant. The authors should stress that the association reported by Kocarnick et al is weak

Line 154: it should be “no significant association have emerged between sex and polymorphisms in other genes”

Line 192: reference 28 is the same of reference 22

Line 196: it should be “data showing”

Lines 250-253: the authors should mention that the observed association derives just from 6 out of 16 works considered in the original review

Lines 264-269: the hypothesis proposed by the author is really weak. Clinically, inflammation is positively correlated with a better clinical outcome in melanoma, as reported by ref. 95 cited by the authors themselves.  Additionally, melanoma microenvironment is strongly immune suppressive and melanoma is one of the cancer types that most benefits from the treatment with anti-immune checkpoint inhibitors, whose function is to reactivate immune response. Thus, considering the apparent contradiction with the clinical evidences, the authors should better address this point, citing other possible explanations as suggested by Song et al (ref. 46)

Lines 332-335: in this paragraph, it is more correct to talk about differences between sexes, not genders

Line 351: refs. 75 and 76 do not support authors’ statement. Lee at al state that “The BRAF mutation was seen in 321 of 798 (40%) male and in 317 of 694 (46%) female patients. The NRAS mutation was observed in 111 of 569 (20%) male and in 110 of 538 (20%) female patients. No relationship was found between sex and BRAF or NRAS mutations (OR = 0.808, 95% CI 0.651–1.002, P = 0.052 and OR = 0.983, 95% CI 0.722–1.339, P = 0.914, respectively)”. Kim et al state that “The incidence of BRAF mutation according to sex was compared in 33 studies. BRAF mutation was seen in 839 of 2325 (36.1%) men and in 770 of 2025 (38.0%) women. No relationship was found between sex and BRAF mutation (OR = 0.879 [95% CI, 0.770- 1.003]; P = .055). There was no statistical heterogeneity among the studies (I2 = 11.560). The subgroup analysis of 27 studies of whites showed statistical significance (OR = 0.854 [95% CI, 0.740-0.986]; P =.032), whereas 6 studies of Asians revealed that there was no significant association between BRAF mutation and sex (OR = 1.022 [95% CI, 0.733-1.424]; P =.898, respectively)”. Thus, a reduced incidence of BRAF mutations in men is reported only by Kim et al and only for whites

Line 389: it should be “found in men”

Lines 395-401: the mechanism proposed by the authors is actually an immunosuppressive pathway. IL-10 is a well-known immunosuppressive cytokine produced by regulatory immune cells as well as melanoma cells. Indeed, the study cited by the authors refers to autoimmune encephalomyelitis. As widely described by, for example, Sabra Klein in several works about the topic, males display weaker immune responses, which make them more susceptible to tumors and infections but less susceptible to autoimmune diseases. The secretion of IL-10 by T cells induced by testosterone showed by Liva et al. can be advantageous in the context of autoimmunity, but it is very unlike that it could occur in the context of melanoma immunotherapy, which is aimed to reactivate, not suppress, immune response. Thus, the authors should better address this point, providing a biological mechanism better fitting with the pathological and clinical features of melanoma.

Line 510: it should be “the” instead of “that”

Author Response

We thank the Reviewer for raising these important issues and appreciate the Reviewer’s suggestions.

Line 70: it should be “by using”

Reply: We replaced “using” with “by using”, as suggested.

Line 111: the authors provided only a Supplementary Table 1 and do not previously mention any other supplementary table. I guess it is a typo, but it should still be amended.

 Reply: We replaced “Supplementary Table 4” with “Supplementary Table 1”, as suggested.

Line 128: it should be “few genes”.

Reply: We replaced “a few” with “few”, as suggested.

Lines 130-131: this sentence is incorrect. If the presence of multiple alleles is documented, as it is, their existence is not a possibility, so the usage of “may” is incorrect in this case. Additionally, a gene carries mutations, not allelic variants. Therefore, the authors should rewrite the sentence amending these errors. If they want to emphasise the presence of non-synonymous mutations, an option could be “since several of its allelic variants show non-synonymous mutations, some of which confer a… etc”.

 Reply: We accepted this comment and rephrased that sentence following the Reviewer’s suggestion.

Line 134: it should be “few studies”.

Reply: We replaced “a few” with “few”, as suggested.

Lines 137-138: I recommend the authors to split the two sentences and related studies. From how it is written, it looks like the two studies have been performed by using the same patients’ cohort, while from the references it is clear that they are two separated studies.

Reply: We split the sentence into two and rephrased them in order to make it clear that the two studies were totally different, as suggested by the Reviewer.

Line 144: ref. 7 only refers to CDKN2A variants. The authors might add it at the end of the paragraph (line 149), since also references n° 8-11 refer to melanoma-prone families.

Reply: Thank you for this comment and suggestion: the reference 7 appears now at the end of the paragraph, alongside references 8-11.

Lines 150-153: The observed SNP is between TPCN2 and MYEOV genes, not within TPCN. It is not a trivial difference, since mutations within the gene affects the function of the protein, while mutations in the flanking regulatory regions generally affect levels of expression. Thus, the text should be amended accordingly. Additionally, Kocarnick et al specify that the association they observed is only marginally associated with CM risk in males, while from the text such association sounds statistically significant. The authors should stress that the association reported by Kocarnick et al is weak.

Reply: We thank the Reviewer for pointing out our inaccuracy in that paragraph. We amended the text accordingly.

Line 154: it should be “no significant association have emerged between sex and polymorphisms in other genes”.

Reply: We modified the text as suggested by the Reviewer.

Line 192: reference 28 is the same of reference 22

Reply: We modified the text as suggested by the Reviewer.

Line 196: it should be “data showing”

Reply: We modified the text as suggested by the Reviewer.

Lines 250-253: the authors should mention that the observed association derives just from 6 out of 16 works considered in the original review

Reply: We modified the text as suggested by the Reviewer.

Lines 264-269: the hypothesis proposed by the author is really weak. Clinically, inflammation is positively correlated with a better clinical outcome in melanoma, as reported by ref. 95 cited by the authors themselves.  Additionally, melanoma microenvironment is strongly immune suppressive and melanoma is one of the cancer types that most benefits from the treatment with anti-immune checkpoint inhibitors, whose function is to reactivate immune response. Thus, considering the apparent contradiction with the clinical evidences, the authors should better address this point, citing other possible explanations as suggested by Song et al (ref. 46).

Reply: We discussed possible explanations by adding the following paragraph encompassing lines: 277-280.

"Another possible mechanism, as discussed in [45], is that smoking can decrease Notch pathway gene expression described to enhance melanoma cells’ growth [47]. Furthermore, smoking increases elastosis, which can reduce the development of melanoma [48]".

Lines 332-335: in this paragraph, it is more correct to talk about differences between sexes, not genders review

Reply: We modified the text as suggested by the Reviewer.

Line 351: refs. 75 and 76 do not support authors’ statement. Lee at al state that “The BRAF mutation was seen in 321 of 798 (40%) male and in 317 of 694 (46%) female patients. The NRAS mutation was observed in 111 of 569 (20%) male and in 110 of 538 (20%) female patients. No relationship was found between sex and BRAF or NRAS mutations (OR = 0.808, 95% CI 0.651–1.002, P = 0.052 and OR = 0.983, 95% CI 0.722–1.339, P = 0.914, respectively)”. Kim et al state that “The incidence of BRAF mutation according to sex was compared in 33 studies. BRAF mutation was seen in 839 of 2325 (36.1%) men and in 770 of 2025 (38.0%) women. No relationship was found between sex and BRAF mutation (OR = 0.879 [95% CI, 0.770- 1.003]; P = .055). There was no statistical heterogeneity among the studies (I2 = 11.560). The subgroup analysis of 27 studies of whites showed statistical significance (OR = 0.854 [95% CI, 0.740-0.986]; P =.032), whereas 6 studies of Asians revealed that there was no significant association between BRAF mutation and sex (OR = 1.022 [95% CI, 0.733-1.424]; P =.898, respectively)”. Thus, a reduced incidence of BRAF mutations in men is reported only by Kim et al and only for whites.

Reply: Once again, we would like to thank this Reviewer for this valuable  comment. We modified the sentence into the following: “Somewhat in contrast with these findings, two-meta-analyses published in 2011 and 2015 did not find any substantial difference between genders in the proportion of BRAF or NRAS mutations (except for a 15% reduced proportion of BRAF mutations among men in white populations, reported by Kim et al.). Likewise, a more recent (2020) systematic review…” etc.

Line 389: it should be “found in men”

Reply: We modified the text as suggested by the Reviewer.

Lines 395-401: the mechanism proposed by the authors is actually an immunosuppressive pathway. IL-10 is a well-known immunosuppressive cytokine produced by regulatory immune cells as well as melanoma cells. Indeed, the study cited by the authors refers to autoimmune encephalomyelitis. As widely described by, for example, Sabra Klein in several works about the topic, males display weaker immune responses, which make them more susceptible to tumors and infections but less susceptible to autoimmune diseases. The secretion of IL-10 by T cells induced by testosterone showed by Liva et al. can be advantageous in the context of autoimmunity, but it is very unlike that it could occur in the context of melanoma immunotherapy, which is aimed to reactivate, not suppress, immune response. Thus, the authors should better address this point, providing a biological mechanism better fitting with the pathological and clinical features of melanoma.

Reply: We thank the Reviewer for highlighting this. We have changed this paragraph. It now reads:

"One hypothesis to explain why among patients treated with ICIs men show slightly greater benefits than women can be found in the sex-related differences of the immune system at baseline, as indicated by the higher number of CD8+ T lymphocytes and the lower CD4+/CD8+ ratio in males than in females [84]. Furthermore, the higher amount of T regulatory cells (Treg) in men, the subpopulation preferentially depleted by anti-CTLA-4 antibodies, could be a factor influencing the different responses, thus producing a more substantial male benefit ([85]. Assuming that women generally have more robust immune systems than men, it is reasonable to expect diverse sex-associated anti-tumor effects of ICIs [86]. Nonetheless, enrolling a balanced number of men and women in clinical trials is of utmost importance to better understand sex differences and achieve the best healthcare for all [87]". 

Line 510: it should be “the” instead of “that”

Reply: We thank the Reviewer for noticing our typo. However, as Reviewer 2 recommended,  we shorten the conclusions and created a limitation section. This particular sentence has been removed.

Reviewer 2 Report

I have read and reviewed with great interest the article entitled "Gender-dependent specificities in cutaneous melanoma predisposition, risk factors, somatic mutations, prognostic and predictive factors: a systematic review".

Attention should be paid to the following points in order to consider the article for possible publication in the journal:

1) The abstract should include the date on which the literature review was conducted.

2) Has the systematic review been registered?

3) How were the studies assessed for risk of bias?

4) A section on the limitations of this work needs to be included in the discussion.

5) The conclusions are too long and there should be no literature references in the conclusions.

6) The bibliographical references are not described according to the journal's guidelines.

Author Response

We appreciate the Reviewer’s comment and we thank the Reviewer for the useful suggestions

1) The abstract should include the date on which the literature review was conducted.

Reply: We modified the text as suggested by the Reviewer.

2) Has the systematic review been registered?

Reply: No, the systematic review was not registered.

3) How were the studies assessed for risk of bias?

Reply: When present, the limitations of each included study were highlighted and discussed, and we specified to interpret the results with caution.

4) A section on the limitations of this work needs to be included in the discussion.

Reply: We modified the text as suggested by the Reviewer: We included a separate limitation section.

5) The conclusions are too long and there should be no literature references in the conclusions.

Reply: We modified the text as suggested by the Reviewer.

6) The bibliographical references are not described according to the journal's guidelines.

Reply: We modified the text as suggested by the Reviewer.

Round 2

Reviewer 2 Report

The authors have made all the necessary modifications. The manuscript is now suitable for publication.